# The Impact of Foreign Direct Investment (FDI) on the Environment: Market Perspectives and Evidence from China

**Jiajia Zheng [1],\* and Pengfei Sheng [2]**

[1]   School of Business, Henan University, Kaifeng 475004, China
[2]   School of Economic, Henan University, Kaifeng 475004, China; sheng_pf@163.com
\*   Correspondence: zjjj-40316266@163.com; Tel.: +86-182-365-73500

**Abstract:** Foreign direct investment (FDI) may have a positive effect on the level of pollution in host countries, as described by the pollution haven hypothesis (PHH). However, this kind of effect may depend on the economic conditions in host countries. In this study, we conduct research on the FDI's effect on China's $CO_2$ emissions during the market-oriented reform. The results are as follows. Firstly, FDI directly promotes China's $CO_2$ emissions. Secondly, with market-oriented reform, this positive effect from FDI is lowering year by year, which indicates that the market-oriented reform could alleviate the positive effect of FDI on China's $CO_2$ emissions. Thirdly, as China's market-oriented reform was implemented gradually from experimental zones to the whole country, regional market development is uneven, and as such so is FDI's effect on local $CO_2$ emissions. Provinces in the eastern area generally evidenced higher market development and lower $CO_2$ emissions from FDI, while four provinces in west area evidenced both lower market development and higher $CO_2$ emissions from FDI.

**Keywords:** FDI; market-oriented reform; $CO_2$ emissions per capita; $CO_2$ emission intensity

## 1. Introduction

Since the introduction of China's "Reform and Opening Up" policy, the Chinese government has put great effort into market-oriented reform and the "go out and bring in" strategy. The amount of foreign direct investment (FDI) that flowed into China grew explosively, and now China has become one of the world´s main assembly grounds. According to the United Nations Conference on Trade and Development (UNCTAD), China brought in more than 100 billion dollars from the outside world in 2014, compared to only 57 million dollars in 1980, during the early years of the reform. As for the total amount of FDI, China is now in second place (behind America) with an average growth rate of 25.49%. There is little disagreement that FDI is an engine and a kind of catalyst for host countries in economic development. However, for those countries in urgent need of economic development, low standards of environmental regulations, cheap resources and energy are always used as incentives to attract foreign investment. Along with economic development, foreign investment brings in a series of environmental problems. For China, the largest developing country in the world, the standards of environmental regulations are still low or even rare in some aspects, compared with those in developed countries. Therefore, whether, and to what extent, the entry of FDI is accompanied by a series of negative environmental problems is an important question yet to be answered.

In this study, we examine the effect of FDI on carbon emissions in China. While China's economy has been growing at unprecedented rates for decades since the market-oriented reform, FDI and

carbon emissions have risen rapidly at the same time. Thus, the market-oriented reform background should not be overlooked in our analysis. Especially since 1992, when the Chinese Communist Party's 14th Congress set up China's economic reform target for building up a socialist market economy, China has quickened its market-oriented steps and pushed reform onto a new stage. Gradually, free competition and a standardized market-price system come to play a leading role in the allocation of most kinds of resources. Simultaneously, China's FDI and carbon emissions experienced explosive growth. The pollution haven hypothesis (PHH) is a famous hypothesis which indicates that developed countries, by investing abroad, often transfer polluting industries or commodities into developing ones, as the latter's environmental regulations are comparatively laxer or even scarce. Then, the developing countries slowly become pollution havens. Under this hypothesis, FDI is the main representative variable in exacerbating the host country's environmental pollution as well as promoting its economy. A sizable amount of literature documents that FDI is associated with environmental pollution in host countries (Friedman et al., 1992 [1]; List and Co, 2000 [2]; Xing and Kolstad, 2002 [3]; Cole, 2004 [4]; He, 2006 [5]; Jorgenson et al., 2007 [6]; Zhang and Fu, 2008 [7]; Baek et al., 2009 [8]; Zeng and Zhao, 2009 [9]; Chung, 2014 [10]). Other empirical findings on PHH are relatively controversial. Cole and Elliott (2003) [11] found that FDI has a negative effect on host country's $CO_2$ and $NO_x$ emissions, while the impact of FDI on $SO_2$ and BOD (Biochemical Oxygen Demand) is positive. Aliyu (2005) [12] made the same conclusions from the data of 11 OECD countries, but for another 14 non-OECD countries, the results were not significant. None of these papers paid any attention to the host countries' objective economic conditions, which may play a decisive role. Hoffman et al., (2005) [13] investigated FDI's effect on host countries' $CO_2$ emissions according to different local income levels. He found that FDI significantly increases the middle-income countries' $CO_2$ emissions, while in high-income countries there is no significant relationship. However, in low-income countries, $CO_2$ emissions can even hinder the entry of FDI. Perkins and Neumayer (2012) [14] investigated 77 countries' dynamic panel data and found that FDI promotes the efficiency of carbon emissions in host countries, and the effect is relevant with local economic institutional factors. Cole and Fredriksson (2009) [15] used a multiple-principal, multiple-agent lobby group model to analyze FDI's effect on host countries' environment. They found that political institutions such as the number of legislative units play a decisive role. Other related factors such as environmental policy, property rights institutions [16], and local human capital level [17] are also included in some empirical papers. Besides analysis from inter-country data, other studies also found stronger evidence in favor of PHH through intra-country data, such as in U.S. (see e.g., Becher and Henderson, 2000 [18]; Morgan and Condliffe, 2009) [19]). Lian et al. (2016) [20] and Zheng et al. (2017) [21] found evidence supporting PHH in China.

Contrary to the PHH, there exists the Porter Hypothesis (PH) which indicates that stricter environmental regulations can encourage enterprises to carry out technical innovation and introduce cleaner energy and/or environment-friendly technology, so countries with stricter environmental regulations would not only affect the relocation of enterprises with FDI, but also make the local environment better. Some studies have affirmed this (see e.g., Birdsall and Wheeler, 1993 [22]; Eskeland and Harrison, 2003 [23]; Kearsley and Riddel, 2010 [24]). They found that cleaner technology and more stringent industry regulations are also introduced by FDI, besides local innovation, so a positive spillover effect for technology and industrial upgrading appears in host countries. After making empirical studies addressing the preferences of enterprises with FDI in China, Di (2007) [25] found that enterprises with FDI are more inclined to settle into provinces with stricter environmental regulations. Dean et al., (2009) [26] views that environment-friendly foreign investment flow is more inclined to use cleaner technology, which would ease the environmental impact on the recipients. From what has been discussed above, technological advances play an important role in the concept of environment pollution from FDI. Using panel data from the Gulf Cooperation Council Countries, Al-mulali and Tang (2013) [27] believed that the PHH does not stand in these countries, FDI promotes economic growth contrarily, so more FDI should be brought in. Using panel of manufacturing sectors of 17 European

countries between 1997and 2009, Rubashkina et al., (2015) [28] investigated the "weak" and "strong" versions of PH; they provide support in favour of "weak" PH, as proxied by patents.

Whether the entry of FDI has increased China's environmental pollution is still unclear, as the studies above give conflicting views. There is one fact that we cannot ignore: the entry of FDI is always correlated with China's market-oriented reform, so the reform background should not be excluded in the analysis of FDI's effect on China's environment. In fact, whether the PHH or the PH stands depends greatly on local economic conditions [17]. Local economic conditions differ among different countries. This is called the heterogeneity, which mainly includes economic development, income levels, factor-market development, legal systems, dynamic relationships between government and market, environmental regulations, non-state-owned economy development, product market development and so on. All of these terms firstly influence the quality and quantity of FDI, and then relate to FDI's effect on local environment. Lin and Du, (2015) [29] analyzed the impact of China's market-oriented reform on China's regional energy and carbon efficiency, and found that the promotion of factor markets (one aspect of market-oriented reform) has a positive effect on efficiency of energy use and $CO_2$ emission. However, such studies are few. This study fills the research gap by introducing China's market-oriented reform, proxied by marketization level in the following analysis of FDI's effect on China's environment, proxied by $CO_2$ emissions.

Our study is arranged as follows. In Section 2, we give a brief introduction of China's market-oriented reform. In Section 3, we describe the methods, variables and data in detail. In Section 4, we empirically measure FDI's effect on China's $CO_2$ emissions through introducing market-oriented reform and present the results. In Section 5, a related discussion is presented. In Section 6, we conclude the paper and provide the policy suggestions.

## 2. China's Market-Oriented Reform

China began its Comprehensive Economic Reform (CER) in 1978, from a central planned economy to a market-oriented economy. Before 1978, the central government controlled macro economy and all production and consumption activities were operated through commands from central to local government. The Five Year Plan was set up to direct the economy (Hou, 2011 [30]). In agricultural production, production teams were organized, and in industrial sectors, state-owned enterprises (SOEs) were organized. As on the means of production such as land, energy and labor, only the government had the right to allocate, according to the economic plan. Moreover, in the supply and demand market, the price was not decided via the interaction of supply and demand, but was set by the government, especially the prices of raw materials and living necessities. Thus, how much people earnt was not in accordance with their contribution, and the egalitarian system of income distribution was established by the government.

Under the planned economy system, the market law was set aside and Chinese economy stagnated for decades. To revive the economy, Chinese government adopted CER and pushed it forward in a gradual way, unlike the shock therapy in Eastern Europe (Gao, 1993 [31]). The first step was implemented in rural areas and the Household Responsibility System (HRS) was set up, under which farmers could make their own production decisions on land usage and obtain the most harvest. The HRS was a great success and China's agricultural production grew dramatically. The second step was in the urban industrial sector, where various "managerial responsibility" systems were introduced in SOEs. Private enterprises and foreign investment were also allowed. With the advancement of the reform, the government gradually gave the right of pricing resources to the market. Moreover, the government's administrative means in managing the economy were replaced by indirect market means and laws. In addition, the government began to act as a watchman for the economy, providing necessary administrative service and public goods only. Since then, prices of most commodities in product market have been determined via the interaction of supply and demand freely and the product competitive market had been basically achieved. In the following step, the factor market was reformed and prices of production factors including capital, energy and land were gradually

determined through the free market. Meng et al., (2016) [32] evaluated how China's market-oriented reform improved the electricity generation efficiency and they found that market competition should be further introduced into China's electric power industry to improve the generation efficiency of the thermal power plants. Lin and Du (2015) [29] analyzed the impact of China's market-oriented reform on regional energy and carbon efficiency. As part of China's market-oriented reform, the "go out and bring in" strategy was paid great attention. With more and more foreign funds and high technology as well as advanced managerial experiences introduced, they came to be an indispensable part in driving China towards modernization.

This CER began with experiment in some specific regions (such as rural areas) and then gradually expanded to the whole country. For the market-oriented reform, eastern area was the first experiment zone, then the successful experiences were spread to the middle area, and finally to the west area. As a result, China's step-by-step market-oriented reform led to unbalanced economic development and regional marketization levels among the areas. For example, provinces in the eastern area are the pioneers of market-oriented reform, as is the economy there. In contrast, provinces in middle areas are less developed and those in the western area are generally the least developed [33].

Fan et al., (2012) [34] performed continuous research on China's provincial market-oriented reform for more than ten years. They measured the development of the entire market, production market, factor market, the relationship between government and market, development of non-state-owned economy, legal environment, and yield data for each province from 1997 to 2009. Figure 1 shows average level of market development for China in these years. A higher score means a higher level of market development. Average carbon emissions and FDI in China at the same period are also shown in Figure 1. We can see that China's FDI and $CO_2$ emissions increased gradually, while the average level of market development also had a similar trend. Therefore, it cannot be ignored in the analysis of FDI's effect on environment. Furthermore, the market development level in every province is different, so the influence on FDI differs, as well as FDI's effect on host country's environment.

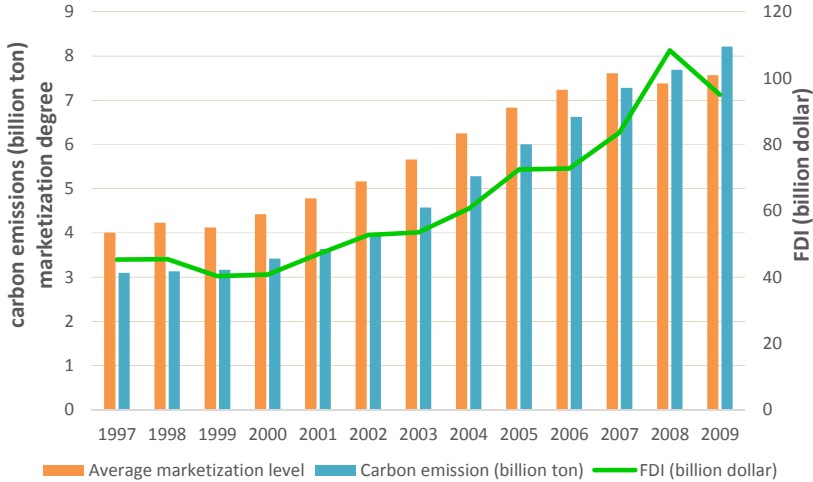

**Figure 1.** China's carbon emissions, foreign direct investment (FDI) and marketization (1997–2009). Source: FDI from the United Nations Conference on Trade and Development (UNCTAD); carbon emissions calculated by author; marketization from Fan et al., (2012) [34].

## 3. Method

### 3.1. Theoretical Model

Grossman and Krueger (1991) [35] first decomposed the impact of economic activities on environment into three main channels: scale effect, structure effect and the technological progress

effect, and these effects have been referenced in many other studies. Accordingly, this article begins with the basic model as following:

$$C = Y \times S \times T \tag{1}$$

In Equation (1), *C* denotes $CO_2$ emission, *Y* denotes economic production, *S* denotes industry structure, and *T* denotes technology.

The economy and the nature environment is a complex organic unity, Färe et al., (2007) [36] named it as "no fire with no smog", which means there will not be any economic production, with no environmental pollution. Taylor et al., (2008) [37] also concluded that production expansion has a negative effect on the natural environment.

The entry of FDI expands the host countries' production scale at first. According to competitive advantage theory, FDI may be invested in what is the host country's comparative advantage, and then on the international market the host may specialize in production of abundance resources endowment, that is to say, FDI expands the host country's production scale and changes its production structure, which is called scale effect and structure effect, respectively. More economic production brings greater energy consumption and environmental pollution at the same time. However, according to the PH, environmental regulations may encourage enterprises to innovate, leading to technology upgrading, which may offset the former negative pollution effect. In usual practice, the FDI's scale effect on environment is indistinguishable from the scale effect of economic production, so in Equation (1), FDI's scale effect on the environment is included in *Y*. On the other hand, regional or national economic structure can also affect $CO_2$ emission (Jorgenson et al., 2007 [6]), and FDI can affect local economic structure, so the variable *S* also includes FDI's structure effect on $CO_2$ emission.

Variable *T* in Equation (1) denotes the technology factor. According to the PH, the host country may improve environmental regulations with its economic development. So, under the pressure of higher standards of environmental regulations, more and more investment of enterprises will be put into research of cleaner technology or production. Besides, clean-technology will also be transferred into local enterprises, as the cooperation and communication between local and foreign enterprises increases. Therefore, FDI's technology effect on the environment is improved. When it offsets the increase of $CO_2$ emissions from FDI's scale and structure effects, we can see that FDI decreases domestic $CO_2$ emissions. However, the technology effect cannot make any sense without local economic conditions. To discuss FDI's effect on local $CO_2$ emissions, economic conditions cannot be ignored. Thus, technology function should be as follows:

$$T = f(market, FDI) \tag{2}$$

Putting Equation (2) into Equation (1), we get the $CO_2$ emission function:

$$C = Y \times S \times T(market, FDI) \tag{3}$$

According to the above, we set our benchmark model as follows:

$$E_{it} = c + \alpha_i + \beta_1 fdi_{it} + \beta_2 market_{it} + \gamma x + \eta_t + \varepsilon_{it} \tag{4}$$

In Equation (4), *i* denotes province, *t* denotes year. $\alpha_i$ denotes areal fixed effect, and $\eta_t$ denotes time fixed effect. *x* denotes vectors for the control variables relating to $CO_2$ emission. $E_{it}$ denotes areal $CO_2$ emission, $fdi_{it}$ denotes the FDI, and $market_{it}$ denotes areal marketization level.

In some studies, market-oriented reform makes a great effort by changing the transaction costs and incentive mechanism, encouraging local research and development activities, which also improves the FDI's technology effect on local $CO_2$ emission, explained by the interaction term of FDI and marketization level. There is also the fact that China's market-oriented reform could change its productive structure, and the productive structure affects its carbon emissions, so FDI's effect should

interact with the productive structure, rather than with marketization level [1]. However, this is not what we emphasize in this paper; what we focus on is whether FDI's effect on China's carbon emissions is influenced by its market-oriented reform. Therefore, in our paper, we pay more attention to the interaction term of FDI and marketization level. As productive structure or industry structure, proxied by the ratio between added value in secondary and tertiary industry, is a nonnegligible factor relating to carbon emissions, we set it as a control variable in our model [2].

To investigate the total effect of FDI on the local environment, the interaction term must be added to the model. Thus, the benchmark model can be written as:

$$E_{it} = c + \alpha_i + \beta_1 fdi_{it} + \beta_2 market_{it} + \beta_3 (fdi \times market)_{it} + \gamma x + \eta_t + \varepsilon_{it} \tag{5}$$

Making a partial derivation of FDI in Equation (5), we get:

$$\frac{\partial(E)}{\partial(fdi)} = \beta_1 + \beta_3 \times market \tag{6}$$

In Formula (6), if $\beta_1 + \beta_3 \times market > 0$, then the marginal effect of FDI on local $CO_2$ emissions will be positive. If $\beta_1 + \beta_3 \times market < 0$, then the marginal effect of FDI on local $CO_2$ emissions will be negative. If $\beta_1 + \beta_3 \times market = 0$, then FDI will have no effect on local $CO_2$ emissions. Moreover, the impact of FDI on local $CO_2$ emission depends not only on $\beta_1$ and $\beta_3$, but also on local marketization level.

### 3.2. Data and Variables

The purpose of this study is to examine whether the market-oriented reform plays a decisive role in FDI's effect on China's $CO_2$ emissions. To implement it, we drew upon provincial market development data from Fan et al., (2012) [34]. The data is continuous from 1997 to 2009, comprehensively describing the market development of 30 Chinese provinces, compared with local income level or the discrete institution variable. We chose $CO_2$ emissions as the environmental variable. China's provincial $CO_2$ emissions data are calculated according to Intergovernmental Panel on Climate Change (IPCC), with both $CO_2$ emission intensity and per capita emission included. We used China as a whole country to investigate FDI's effect on recipient's environment, eliminating the interference of heterogeneity among different countries. All the variables are described as following.

3.2.1. The Dependent Variable: Carbon Emission

There are no official statistics of China's provincial carbon emission data, so we calculated the data according to "Guidelines for National Greenhouse Gas Inventories" (Chapter 6, Volume 2) [38]; the computation formula is as follows:

$$C_t = \sum_{i=1}^{3} C_{it} = \sum_{i=1}^{3} E_{it} \times NCV_i \times CEF_i \times COF_i \times (44/12) \tag{7}$$

where $C$ is $CO_2$ emissions, $i = 1,2,3$ denotes three main $CO_2$-emitting energy: coal, petrol and gas, $t$ denotes year, $E$ is the energy consumed, $NCV$ is net calorific value, $CEF$ is coefficient of carbon emissions, $COF$ is carbon oxidation factor, 44 and 12 are chemical molecular weight of $CO_2$ and $C$,

---

[1]　An editor gave this comment, and we thank him/her very much, because this comment improved our research framework and made the paper more constructive and convincing.

[2]　Actually, we performed more regression by adding the interaction term of FDI (proxied by "*fdi*" in data) and productive structure/industry structure (proxied by "*in23*" in data) into the benchmark model. The results show that the coefficient is insignificantly negative, especially after we added the interaction term of "*fdi*" and "*marketization level*". For lack of space, we did not display the results here. If anyone is interested, please email us.

respectively. In determining the *CEF*, we referred to Chen (2011) [39], and the results were 2.763 kg $CO_2$ per standard for coal, 2.145 for petrol, and 1.642 for gas. According to the Chinese Energy Yearbook and Equation (7), we calculated $CO_2$ emission data for 30 provinces from 1997 to 2009. Considering the regional differences in economic development and population, using gross domestic product (GDP) and population as weights, we obtained the final dependent variables: $CO_2$ emission intensity and $CO_2$ emission per capita, presented as gco2 and pco2 in the following, respectively.

### 3.2.2. Core Independent Variables: Marketization Level and FDI

At a marketization level, data is from Fan et al., (2012) [38]. With respect to FDI, we found the volume of foreign investment put into actual use of the 30 provinces in China from 1997 to 2009. To eliminate the difference among provinces, we divided the volume of FDI by the volume of local fixed assets investment every year. Finally, the ratio is used to describe the use of FDI in each province.

### 3.2.3. Control Variables

In this study, we also chose other variables related to carbon emissions as follows:

(1) GDP per capita (abbreviated as pgdp): the ratio of GDP and local population every year. All GDP data is deflated into the 1995 constant price.

(2) The ratio of added value in secondary and tertiary industries (abbreviated as in 23): this ratio is to measure the effect of local industry structure on $CO_2$ emissions. The secondary industry especially needs a large volume of energy and emits a lot of $CO_2$, so the higher the ratio is, the more the local economy depends on the secondary industry, and the higher local $CO_2$ emission is.

(3) Energy intensity (abbreviated as ei): the ratio of local total energy consumption and the GDP. Ever since the adoption of the "reform and opening up" policy, and along with industrialization and urbanization, energy demand, especially for coal, has grown sharply, along with more and more $CO_2$ emission.

(4) Carbon emissions from neighbourhood (abbreviated as yl): In contrast with other types of environmental pollution, such as $SO_2$, NOx and waste water, there is no official data on carbon emission regulations or capital into carbon-reduction. Therefore, the environmental regulation level cannot be described easily. Against the background of "Energy Conservation and Emission Reduction", the regional government is inter-related and competitive with each other as well, thus the $CO_2$ emission coming from a neighbourhood may be an invisible measurement that local government uses to set its own emission-reduction indicators. In other words, the $CO_2$ emissions from a neighbourhood affect local $CO_2$ emissions indirectly. Here, we use information of whether the two provinces have the same land boundary as an indicator to define 'neighbourhood', calculating $CO_2$ emission intensity from all neighbourhoods, then consider it as an invisible pressure for local government. Actually, we set this variable as an environmental regulation indicator.

(5) Unemployment rate (abbreviated as unem): the yearly registered unemployment rate in statistical yearbook. When the unemployment rate rises, the local government will not be promoted by the central government. As a result, promoting employment of residents is an important duty of local government. Moreover, a comparative rise in the unemployment rate would force local government to turn a blind eye toward introducing some new industries or projects, and then environmental regulations might be set aside. As the unemployment rates rise, the local government will pay more attention to residents obtaining work and being promoted rather than energy conservation and emission reduction technology.

Data of all the variables above have been taken from the Chinese Statistical Yearbook, the Chinese Energy Statistical Yearbook, the Assemble Statistical Data of New China 60 Years (1949–2009), the statistical database on CCER, and the Chinese Economic Research Network.

## 4. Empirical Results

### 4.1. Unit Roots Testing for Panel Data

As we know well, more and more empirical literature has recently focused on the stationarity of variables in panel data as an extension of the empirical estimation method. Thus, testing for unit roots in heterogeneous panels, so as to confirm the stationarity of all variables in our panel data, should come first. Econometric theory has developed many ways, such as the LLC test (Levin et al., 2002 [40]), HT-test (Harris and Tzavalis, 1999 [41]), Breitung-test (Breitung and Hassler 2002 [42]), IPS-test (Im et al., 2003, [43]), Fisher-test (Choi and Ahn, 1999 [44]), Hadri LM-test (Hadri, 1996 [45]). Different methods have different assumptions and scope of application. Among these methods, the LLC test, the HT test and the Breitung test assume a common root while the other methods do not. Meanwhile, time series and cross-section series are contained simultaneously in panel data, so the most applicable method lies in the sample size of panel data.

In this paper, the variables come from 30 provinces in China from 1997 to 2009, so time dimension $T = 13$ while cross-section dimension $N = 30$, and this is a short panel data. According to the applicable sample size of all the above methods, we choose the HT test, the IPS test and the Fisher test here to do unit roots test. The testing results are in Table 1 as following:

**Table 1.** Testing results for unit roots in panel data.

| Variable | HT-Test | IPS-Test | Fisher-Test |
|---|---|---|---|
| | **H0: Panels Contain unit Roots Ha: Panels Are Stationary** | **H0:All Panels Contain Unit Roots Ha: Some Panels are Stationary** | **H0:All Panels Contain unit Roots Ha: At Least One Panel Is Stationary** |
| *pco2* | −0.3561(0.3609) | −2.1256(0.0168) | 5.8239(0.0000) |
| *gco2* | −3.0179(0.0013) | −2.2643(0.0118) | 7.1931(0.0000) |
| *market* | 0.6401(0.7389) | −2.3930(0.0084) | 11.6965(0.0000) |
| *fdi* | −2.0214(0.0216) | −3.0478(0.0012) | 3.8969(0.0000) |
| *market × fdi* | −2.7730(0.0028) | −31.9189(0.0000) | 7.4689(0.0000) |
| *yl* | −3.5846(0.0002) | −2.9259(0.0017) | 6.9743(0.0000) |
| *ei* | −1.5047(0.0662) | −3.6093(0.0002) | 7.2924(0.0000) |
| *in23* | −1.2352(0.1084) | 0.3272(0.6282) | 8.0885(0.0000) |
| *unem* | −3.5668(0.0002) | −5.3008(0.0000) | 13.7690(0.0000) |
| *pgdp* | −8.3889(0.0000) | −6.8728(0.0000) | 3.8487(0.0000) |

Source: Calculated by Stata 11.0. Note: in the HT test, the *z* statistical value is given; in the IPS test, the *z-t*-tilde-bar statistical value is given above; in the Fisher-test, all of the statistical values of inverse chi-squared, inverse normal, inverse logit *t* and modified inverse chi-squared are given, but here only the value of the last one is given; the ( ) after the statistical value is the corresponding *p*-value.

In Table 1, we can find that: (1) under the Fisher test, the assumption H0 of all variables is rejected, which means that for each variable, there is at least one stationary panel; (2) under the IPS test, the assumption H0 is also rejected expect the '*in23*' variable, which means that there are some stationary panels in variables expected '*in23*'; (3) under the HT test, the assumption H0 of three variables (pco2, market, in23) is accepted, which means that in these three variables, units roots may exist; unlike the alternative hypothesis (Ha) of the other two tests, the Ha in the HT test means that all the panels of the tested variable are stationary, so the rejected H0 in the HT test for the rest variables names that these variables are all stationary. In conclusion, most variables in our panel data are stationary while the stationarity of variables about 'market', 'pco2' and 'in23' seems controversial. In light of the different assumption Ha of the three tests, we know that a few panels in each of the three variables are stationary, and the rest contain unit roots, which indicates they are nonstationary. Since the time dimension is just 13 years, which is quite short for an ideal cointegration analysis (the usual estimation method for nonstationary series), and the nonstationary panels are so comparatively small that they would not do any influence on the estimation results, here we try to overlook it and assume that all variables contain no unit roots. We obtained the following empirical result.

### 4.2. Benchmark Model Regression Results

Table 2 shows the regression results according to benchmark model in Equation (5).

**Table 2.** Benchmark model regression results.

| variables | OLS | | Fixed Effect | Random Effect |
|---|---|---|---|---|
| | **pco2** | **gco2** | **pco2** | **gco2** |
| *market* | 0.693 *** | 6.761 *** | 0.787 *** | 6.761 *** |
| | (8.26) | (1.89) | (9.14) | (6.62) |
| *fdi* | 28.677 *** | 73.669 ** | 11.980 *** | 73.669 * |
| | (7.56) | (7.85) | (9.37) | (1.79) |
| *market × fdi* | −4.703 *** | −19.257 *** | −2.785 *** | −19.257 *** |
| | (−7.61) | (−3.04) | (−6.68) | (−2.80) |
| *pgdp* | 0.000 *** | 0.000 | 0.000 ** | 0.000 |
| | (18.05) | (0.29) | (2.53) | (0.21) |
| *yl* | 0.499 *** | −0.338 | −0.604 *** | −0.338 |
| | (7.22) | (−0.48) | (−3.27) | (−0.39) |
| *ei* | 1.364 *** | 15.345 *** | 1.145 *** | 15.345 *** |
| | (12.84) | (14.07) | (5.82) | (8.57) |
| *in23* | 0.686 *** | 12.352 *** | 2.009 *** | 12.352 *** |
| | (2.41) | (4.24) | (5.31) | (3.87) |
| *unem* | 0.000 | 0.557 | 0.159 | 0.557 |
| | (0.00) | (0.57) | (1.65) | (0.54) |
| *_cons* | −7.930 *** | −42.074 *** | −2.711 ** | −42.074 ** |
| | (−11.30) | (−5.84) | (−2.17) | (−4.40) |
| *N* | 390 | 390 | 390 | 390 |
| $R^2$ | 0.740 | 0.500 | 0.710 | – |
| Hausman Test | – | – | 125.03 [0.000] | 17.40 [0.0150] |
| Heteroscedasticity Test | – | – | 8679.19 [0.000] | – |
| Serial Correlatin Test | – | – | 125.94 [0.000] | 22.944 [0.000] |

Source: calculated by Stata 11.0. Note: ( ) is the *t*-value; [ ] is corresponding *p*-value. * $p < 0.10$, ** $p < 0.05$, *** $p < 0.01$. Heteroscedasticity and serial correlation are considered at the same.

As all variables are stationary, we can estimate them through OLS (Ordinary Least Square) and relative ways. Firstly, we used the least squares method and the results are shown in columns (1) and (2). However, for panel data, the OLS estimate assumes all individuals hold for an identical regression equation, thus heterogeneity among different individuals is omitted, leading to inconsistent estimation results. On the basis of OLS, we chose panel fixed effect regression and random effect regression, holding the individual effect to estimate the model. The Hausman test was used here to evaluate which one is more suitable for the panel data, whose null hypothesis is the fixed effect regression and is more suitable. From the *p*-value of the Hausman test, when the dependent variable is pco2, fixed effect is more suitable, when gco2 is the dependent variable, random effect is more suitable. Secondly, in static panel regression, heteroscedasticity and a serial correlation test are essential to the consistence of estimation results. In our benchmark model, after considering heteroscedasticity and serial correlation, the results are in columns (3) and (4).

According to the regression results in Table 2, FDI has a significantly direct positive effect on both pco2 and gco2, which means with the entry of FDI, China's $CO_2$ emission clearly increases. Since the

adoption of the "reform and opening up" policy, inexpensive Chinese labor and energy resources, as well as laxer environmental regulations are more attractive for foreign investment. Moreover, the Chinese local government promotion mechanism, characterized by "GDP first, promotion first", encourages local government to pay close attention on the economic effect from FDI, rather than the environmental effect. Thus results can be easily anticipated, more and more high-energy consumption or high-pollution emission projects are established during this period. As to the impact of market-oriented reform on $CO_2$ emission, the results are also significantly positive. Along with the market-oriented progress, market price plays a decisive role in allocating resources. The usage efficiency of most resources and productivity rise greatly in a short time. On the background of pollution-control technology remaining static, the direct result is expansion of the economy scale, leading to the same increase in $CO_2$ emission. In contrast, the interaction terms of FDI and marketization in columns (3) and (4) are both significantly negative, which is quite interesting.

On the surface, the interaction term signifies that FDI indirectly affect China's $CO_2$ emission through marketization. With the upgrading of areal market-oriented reform, the degree of FDI that directly increases local $CO_2$ emission is lowering year by year, which indicates that the market-oriented progress is making a mitigation effort on FDI's positive effect on China's $CO_2$ emission. As to the causes, in combination with the indicators included in market-oriented reform, the higher the local marketization level is, the more mature the factor-market and product-market are, and the more improved the legal and financial mechanisms are. Therefore, there is always an abundance of skillful human resources and technical capital when the public demands it, and these are the main factors for FDI to produce its technical effect, which will not only promote local economy, but will also improve local environmental quality by increasing the efficiency of energy usage through production technology upgrading, then through $CO_2$ emission decreases.

In control variables, the coefficient of energy intensity (ei) is significantly positive, which denotes that the higher the energy intensity, the greater the $CO_2$ emissions. The coefficient of industrial structure is positive too, which denotes that the higher the added value in the secondary industry, which heavily relies on fossil fuels, the greater the $CO_2$ emissions. Moreover, coefficients of unemployment rate (unem) and emission pressure from the neighborhood (yl) are both insignificant, which are related to the variable selection or model set.

### 4.3. Improved Model Regression Results

In economic activities, the variables are correlated with each other. In this study, provinces with high $CO_2$ emissions are always those with low efficiency of energy utilization and technology. Thus, the need for government intervention in these provinces is more urgent—in order to promote the local economy—so more and more FDI would be invested into high-emission industries or more and more FDI with high emissions would be introduced. In other words, FDI and local $CO_2$ emission influence each other. This is known as an endogenous relationship in econometric analysis, and it has attracted the attention of many scholars. Without analyzing the endogenous relation between $CO_2$ emissions and FDI, the above results are unreliable.

We made an effort to overcome the endogenous relation from the following two aspects:

(1)  Considering the lag effect of all variables. Given the intrinsic connecting economic system, the impact of FDI and marketization level on $CO_2$ emission may continue from this year to the next year, but cannot be turned over from next year to this year. So, here we used the one-year lagged independent variables, instead of the original ones, in an improved model. Using lagged form has two advantages here, firstly to reduce the endogenous degree in current variables and secondly, to reduce the difficulty in searching for appropriate instrument variables. The results are in columns (1) and (2) in Table 3.

**Table 3.** Improved model regression results.

| Variables | XTIVREG | | DIFF-GMM | | SYS-GMM | |
|---|---|---|---|---|---|---|
| | **pco2** | **gco2** | **pco2** | **gco2** | **pco2** | **gco2** |
| $pco2_{t-1}$ | – | – | 0.767 *** (50.74) | – | 0.986 *** (29.60) | – |
| $gco2_{t-1}$ | – | – | – | 0.391 *** (11.51) | – | 0.965 *** (18.97) |
| *market* | 0.893 *** (8.46) | 6.207 *** (7.13) | 0.323 *** (6.35) | 6.007 *** (7.79) | 0.139 *** (3.33) | 0.984 ** (2.77) |
| *fdi* | 18.164 *** (3.65) | 183.981 *** (4.48) | 3.949 (1.37) | 191.279 *** (3.75) | 2.405 (1.45) | 31.124 *** (3.04) |
| *market* × *fdi* | −4.191 *** (−4.72) | −44.248 *** (−6.04) | −0.705 * (−1.66) | −33.729 *** (−4.20) | −0.319 (−1.36) | −5.467 *** (−3.15) |
| *pgdp* | 0.000 ** (2.36) | 0.000 * (1.68) | 0.000 *** (2.69) | 0.000 (0.87) | 0.000 (−0.14) | 0.000 (0.25) |
| *yl* | −0.490 *** (−3.30) | −4.457 *** (−3.74) | −0.137 *** (−4.54) | −4.413 *** (−6.87) | −0.054 (1.52) | −0.292 (−1.12) |
| *ei* | 1.337 *** (6.91) | 13.260 *** (8.31) | 0.879 *** (20.02) | 6.096 *** (6.04) | 0.194 ** (2.45) | 1.817 ** (2.21) |
| *in23* | 1.898 *** (4.22) | 9.059 ** (2.44) | 0.055 (0.31) | −4.197 * (−1.73) | 0.064 (0.62) | 1.914 (1.34) |
| *unem* | 0.292 ** (2.13) | 4.592 *** (4.06) | 0.266 *** (6.70) | 2.777 *** (3.75) | 0.106 ** (2.69) | 0.482 (1.44) |
| *_cons* | – | – | −2.803 *** (−10.93) | −9.323 (−1.54) | −1.538 *** (−3.30) | −7.809 ** (−2.77) |
| N | 360 | 360 | 360 | 360 | 360 | 360 |
| AR(1) | – | – | −3.051 [0.002] | −3.489 [0.001] | −3.00 [0.003] | −3.98 [0.000] |
| AR(2) | – | – | −1.199 [0.230] | −1.637 [0.102] | −1.06 [0.288] | −1.20 [0.232] |
| Sargan test | 1.14 [0.767] | 3.72 [0.294] | 23.92 [1.000] | 22.78 [1.000] | 20.15 # [1.000] | 25.42 # [1.000] |

Source: calculated by the author. Note: ( ) is the t-value and [ ] is the corresponding *p*-value.* $p < 0.1$, ** $p < 0.05$, *** $p < 0.01$. AR is the abbreviation of auto-correlation, which is to test the auto-correlation of the residual. AR(1) refers to the first-order residual auto-correlation, while AR(2) refers to the second-order residual auto-correlation. The null hypothesis of the AR test is that the auto-correlation of the residual occurs. The Sargan value is to test whether the instrumental variables are identifiable appropriately. The null hypothesis here is that the instrumental variables are appropriately identified. # Here gives the value of Hansen test for instruments.

(2)  Considering the sustainability of the impact. The current dependent variable—$CO_2$ emissions are affected by the current independent variables, and by the lagged variables in the last year. To include the impact from the last year, we added the one-year lagged $CO_2$ emission variable in the model. Here is the new improved model, which is also a dynamic panel model:

$$E_{it} = c + \alpha_i + \beta_0 E_{i,t-1} + \beta_1 fdi_{it} + \beta_2 market_{it} + \beta_3 (market \times fdi)_{it} + \gamma x + \eta_t + \varepsilon_{it} \qquad (8)$$

The addition of the one-year lagged dependent variable can estimate the inertial trend, but the endogenous problem cannot be ignored either. At present, there are two solutions to the endogenous problem in the dynamic panel model—the difference of GMM (abbreviated as DIFF-GMM) and the system of GMM (abbreviated as SYS-GMM). For the DIFF-GMM method, the difference of the original equation is estimated to eliminate the areal fixed effect, and the lagged terms of the independent

variables are used instead of the current ones as the instrument variables. For the SYS-GMM method, both the difference equation and the level equation are estimated. Moreover, in the SYS-GMM method, the lagged terms of the difference variables are used in level equation as instruments. Thus, compared with the DIFF-GMM method, there are more instrument variables used in the SYS-GMM method, which will enhance the accuracy of the estimation. In both methods, difference variables and lagged variables are used as instruments, so serial correlation and excessive identification tests should be done. In Table 3, columns (3)–(6) show the results, respectively. And the test results for serial correlation and excessive identification for instruments are also given.

In Table 3, whatever method is used, coefficients of core independent variables remain consistent with benchmark model regression—the coefficient of FDI and marketization level is significantly positive while the coefficient of the interaction term is significantly negative. As the endogenous problem and persistence of all the variables are controlled by instrumental variables and dynamic lagged ones, the results of the improved model are robust and credible. From the results in Table 3, FDI positively increased China's $CO_2$ emissions, and the increasing effect of FDI on China's $CO_2$ emission intensity is a few times than that of China's $CO_2$ emission per capita. However, with the local economic conditions, the interaction term of FDI and marketization level is obviously negative, which means that the increasing impact of FDI on China's $CO_2$ emissions is lessening gradually with the market-oriented reform. As discussed above, market-oriented reform improves the factor-market and product-market, the enforcement efficiency of local government, the legal system mechanism, and the environmental regulations, and under these improvements, the entry of FDI could be screened ever more strictly, so that only clean and environment-friendly FDI can be introduced, especially in areas with high marketization levels. Furthermore, the first aim in introducing FDI is to improve the local economy, so FDI's effect on China's $CO_2$ emissions is firstly on the $CO_2$ emission intensity specifically. That is the reason why the coefficient of FDI on $CO_2$ emission intensity is much larger than that on $CO_2$ emission per capita; the latter is assumed as the $CO_2$ emission by consumer population.

## 5. Further Discussion

Based on the SYS-GMM method, the average direct effect coefficient of FDI on $CO_2$ emission per capita is 2.405, while on $CO_2$ emission intensity, it is 31.124. The average indirect effect coefficient of FDI on both is −0.319 and −5.467, respectively. We set Formula (6) = 0, and calculated the critical value of marketization level, which is 7.539 for $CO_2$ emission per capita, and 6.024 for $CO_2$ emission intensity. Using local marketization level data in 2009, we obtained a closer look on how FDI and $CO_2$ emissions perform in each province:

(1)  Where the marketization level is above 7.539: Tianjin, Hebei, Shanxi, Liaoning, Jilin, Shanghai, Jiangsu, Zhejiang, Anhui, Fujian, Jiangxi, Shandong, Henan, Hubei, Guangdong, Chongqing, Hunan, Sichuan. In these 18 provinces, the total marginal effect of FDI on $CO_2$ emission is negative, which means that local $CO_2$ emissions decrease with the entry of FDI.

(2)  Where the marketizaiton level is above 6.024 but under 7.539: Beijing, Shaanxi, Xinjiang, Hainan, Guizhou, Gansu, Heilongjiang, Inner Mongolia. In these 11 provinces, the total marginal effect of FDI on $CO_2$ emissions is still unclear, as FDI has a negative effect on $CO_2$ emission intensity but a positive effect on $CO_2$ emissions per capita. It is in these areas that more attention should be paid to the reduction of $CO_2$ emission per capita and the market-oriented reform.

(3)  Where the marketization level is under 6.024: Guangxi, Yunnan, Qinghai, Ningxia. In these 4 provinces, the total marginal effect of FDI on $CO_2$ emission is clearly positive. Thus, these areas are the first and most important provinces for Chinese government to introduce an "energy-saving and emission-reduction" policy. Putting great effort into local market-oriented reform would also be a strategy.

## 6. Conclusions

In previous studies of FDI's effect on the recipient's environment, the most attention was paid to the causality of the two and the composition of the environment effect from FDI. The results are not always consistent with each other. Thus, whether PHH occurs is still a mystery. Furthermore, using panel data across countries ignores heterogeneity among different countries, especially the developed and developing ones, so the results may be incredible with no practical economic significance. Only a few studies have made an effort to include the recipient's capital-market conditions, legal mechanisms, and patent and technical conditions to analyze FDI's effect. However, using one or some of these aspects cannot comprehensively describe the recipient's economic conditions. With market-oriented progress, the market itself makes a great effort in screening potential FDI and promotes its technology-upgrading effect in the host countries. Actually, market-oriented reform plays the role of a catalyst for FDI's effect on the recipient and that is what we have done in this study.

From the perspective of market-oriented reform, this study collects panel data for 30 provinces in China, from 1997 to 2009, to perform an empirical test on FDI's effect on China's $CO_2$ emissions. The results show that in China, FDI increases local $CO_2$ emissions directly, but the increasing effect lowers with the upgrading of market-oriented reform. The interaction term of FDI and marketization level is significantly negative, so the higher the local marketization level is, the lower $CO_2$ emissions from FDI are. To some extent, the negative effect may surpass the direct positive effect and then FDI would decrease local $CO_2$ emission comprehensively. Then, based on the SYS-GMM regression results, we calculated the critical value of marketization level and classified the 30 provinces according to the performance of FDI and $CO_2$ emission.

Some related questions may warrant further research. We estimate only the comprehensive effect of provincial marketization level on FDI's effect on China's $CO_2$ emissions. It would be interesting to estimate how the sub-aspects in market-oriented reform perform on China's $CO_2$ emissions from FDI. In addition, we only focus on China's $CO_2$ emissions here, so the next stage would be to test FDI's effect on other types of pollution in China and to compare the differences. Empirical testing of FDI's effect on the recipient's environment would be a very interesting and practical topic in the near future because environmental pollution is becoming more and more important for all countries in the world, sometimes surpassing the importance of economic development. Therefore, how to treat FDI and induce it to play an appropriate role in local economy and in the environment as well becomes greatly urgent.

**Author Contributions:** The paper is a joint contribution of two authors.

**Conflicts of Interest:** The authors declare no conflict of interest.

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
