# Peer review of "The Impact of Foreign Direct Investment (FDI) on the Environment: Market Perspectives and Evidence from China"

_economies, doi:10.3390/economies5010008_

Round 1

Reviewer 1 Report

The paper is in general well-written and methodologically rigorous. After minor revisions, it can be published. I only have two rather minor suggestions. First, the literature review part should be better structured.The author(s) discusses relevant literature in both the introduction and methodology (background) sections, which causes confusion. In addition, the discussion of relevant literature is not thorough enough and should be improved by bringing more recent studies -- not just about China. Second, the writing is in general fairly good, but still needs further editing. Indeed, there is much awkwardness throughout. For example, "economy condition" (abstract) should be "economic condition(s);" "the Chinese government put" should be "the Chinese government has put." The author(s) should hire an editor before resubmitting the paper.

Author Response

Dear Professor:

      Many thanks for your email of 14 Dec 2016, regarding the revision and advice of the above paper in Journal of Economies. Overall the comments have been fair, encouraging and constructive. We have learned much from it. 

      After carefully studying  your advice, we have made corresponding changes to the paper as following:

      1. we improved the introduction and added some relevant referrences

      2. we corrected spell errors all through the paper

      3. comments from another professor required tesing for unit roots, reconstruct the research design, and clearly present the emprirical methods and results, and we have made corresponding changes in the paper.

      Besides the above changes, we have corrected some expression errors. And all the corrections are written in red in order to find them expediently. Hope these will make it more acceptable for publication. 

      The revised manuscript  is attached. If you have any question about this paper, please don’t hesitate to let me know.

       Yours sincerely

Reviewer 2 Report

As the authors know very well, a large body of empirical literature has recently considered the nonstationarity of the panel data as an additional extension of the estimation of the PHH. For example, if the variables used in Equation (4) (p. 5) are nonstationary, cointegration analysis would be more appropriate than other panel estimation techniques such as the fixed/random effects model used for this study (due to statistical inferences). To validate the empirical findings in this paper, therefore, before estimating the model, the authors need to conduct unit root tests. If all the variables are found to be nonstationary, the authors should adopt panel cointegration techniques, rather than the fixed/random effects.      

References:

Im, K.S., Pesaran, M.H., Shin, Y., 1997. Testing for unit roots in heterogeneous panels. Mimeo, Department of Applied Economics, University of Cambridge. Available on site, http://www.econ.cam.ac.uk/faculty/pesaran/lm.pdf

Kao, C., Chiang, M., 2000. On the estimation and inference of a cointegrated regression in panel data. In: Baltagi, B. (Ed), Nonstationary Panels, Panel Cointegration and Dynamic Panels.

Pedroni, P., 1999. Critical values for cointegration tests in heterogeneous panels with multiple regressors, Oxford Bulletin of Economics and Statistics 61(S1), 653-670.

Author Response

Dear Professor:

     Many thanks for your email of 17 Jan 2017, regarding the revision and advice of the above paper in Journal of Economies. Overall the comments have been fair, encouraging and constructive. We have learned much from it. 

     After carefully studying the reviewer’ comments and your advice, we have made corresponding changes to the paper. Major changes include:

     1. we reconstruct the research design and make it more appropriate.

      2. we add test of unit roots before empirical analysis. 

      3. we improve the presentation of research method and the empirical results

      4. the conclusion is also revised

      Besides, we corrected spell errors and expression errors all through the paper.

      The manuscript revised is attached. The changes are written in red so as to find them easily.

      Hope this revision will be more applicable for publication.

      Yours Sincerely

Round 2

Reviewer 2 Report

The authors have done an excellent job on revising the initial manuscript.

Author Response

Dear prof.

     Thank you for your comments on our paper. 

     As you have pointed out in the comments, we have carefully read our paper, checked out all language errors and revised them. 

     We hope our revision version would be more convincing. 

     Best wishes!

                       yours sincerely
